# Extracellular Vesicles and Their Role in Lung Infections

**DOI:** 10.3390/ijms242216139

**Published:** 2023-11-09

**Authors:** Shadi Hambo, Hani Harb

**Affiliations:** Institute for Medical Microbiology and Virology, University Hospital Dresden, Technical University Dresden, Fetscherstr. 74, 01307 Dresden, Germany; shadi.hambo@ukdd.de

**Keywords:** extracellular vesicles, exosomes, EVs, lung infection, COVID-19, bacterial infection, biomarker

## Abstract

Lung infections are one of the most common causes of death and morbidity worldwide. Both bacterial and viral lung infections cause a vast number of infections with varying severities. Extracellular vesicles (EVs) produced by different cells due to infection in the lung have the ability to modify the immune system, leading to either better immune response or worsening of the disease. It has been shown that both bacteria and viruses have the ability to produce their EVs and stimulate the immune system for that. In this review, we investigate topics from EV biogenesis and types of EVs to lung bacterial and viral infections caused by various bacterial species. *Mycobacterium tuberculosis*, *Staphylococcus aureus,* and *Streptococcus pneumoniae* infections are covered intensively in this review. Moreover, various viral lung infections, including SARS-CoV-2 infections, have been depicted extensively. In this review, we focus on eukaryotic-cell-derived EVs as an important component of disease pathogenesis. Finally, this review holds high novelty in its findings and literature review. It represents the first time to cover all different information on immune-cell-derived EVs in both bacterial and viral lung infections.

## 1. Introduction

Extracellular vesicles (EVs) are eukaryotic- and prokaryotic-cell-derived heterogeneous populations of lipid-bilayer-membrane-enclosed structures, which are released in the extracellular space [1,2,3]. Eukaryotic-cell-derived EVs are of high importance in disease pathogens. The release of vesicles by cells into the extracellular matrix was first described as a method of discarding unneeded materials almost forty years ago [4,5]. However, it has been known that together with receptor ligands, signaling molecules, and hormones, EVs are necessary for cell–cell communication and interaction to achieve the coordination of almost all physiologic and metabolic processes [2]. Over the years, there have been multiple classifications for EVs [6,7,8,9,10]. Bacterial EVs, on the one hand, had their distinctive nomenclature: outer membrane vesicles (OMV) in gram-negative bacteria and extracellular vesicles or membrane vesicles (EVs or MVs) in gram-positive bacteria [11]. On the other hand, in eukaryotic cells, EV experts have now classified EVs into two main distinct groups:

Ectosomes (also termed microvesicles, microparticles, or oncosomes), range between 50–10,000 nm in diameter, are formed and released by the outward budding of the plasma membrane [12,13]. Particles of this group possess special markers, such as annexin A1 and ADP-ribosylation factor 6 (ARF6) [9,13]. Since the ectosomes are formed by the outward budding of the plasma membrane, they show a similar composition apart from that the lipid composition among the ectosomes membrane bilayer is evenly distributed compared to the asymmetrical distribution of the lipids in the two leaflets of the plasma membrane [14,15,16]. The protein content of the ectosome’s membrane bilayer and cargo is similar to the plasma membrane and the cytosolic proteins of the producing cell. However, due to the formation by budding, ectosomes lack proteins directly associated with organelles such as the nucleus, Golgi apparatus, and the endoplasmic reticulum, since these organelles do not contribute to the biogenesis of ectosomes [17,18]. Interestingly, however, mitochondrial content has been discovered within the ectosomes [19].

Exosomes 30–150 nm in diameter are produced through the endocytic pathway, where the endosomal membrane experiences an inward buddying that results in what is called intraluminal vesicles (ILVs) [1]. ILVs resemble the pre-released exosomes. Following the formation of ILVs, the endosome is now called a multivesicular endosome (MVE), which releases its content of exosomes to the pericellular matrix after fusion with the plasma membrane [20]. Although the available isolation methodologies and techniques permit the classification of EVs based on surface antigen, density, and size, these strategies cannot determine the site of biogenesis of these EVs, making the identification of the two groups (ectosomes and exosomes) challenging due to the overlap between their particles’ composition and size [21]. Like ectosomes, exosomes have several distinguishing markers, such as the endosomal complexes required for transport (ESCRT): Alix, flotillin, TSG101, and Ras-associated binding protein (Rab) 5b proteins and tetraspanin members CD63, CD9, CD81, and CD82 [7,22,23]. Additionally, exosomes carry the cytosolic proteins Rabs (Ras-associated binding proteins), which participate in promoting exosome docking and membrane fusion [24,25], and annexins, a protein family whose members are presumed to regulate the dynamics of the membrane cytoskeleton and membrane fusion events [25]. The exosomes also carry lipids and metabolites [26,27] as well as a nucleic acid cargo, which is included among the functionally active exosomes when fused with the recipient cells. This may involve a range of non-coding RNAs, like miRNA; long non-coding RNA (lncRNA); fragments of tRNA; structural RNAs; small interfering RNAs; small RNA transcripts; and RNA-protein complexes. In addition to the different RNA species, exosome cargo contains DNA that could represent the whole genome and the genomic mutations [28,29,30,31] as well as mitochondrial DNA [32,33] (Figure 1).

Moreover, other EV subtypes, such as apoptotic bodies (50–5000), that are released by apoptotic cells have been described. They carry phosphatidylserine as a unique marker [34]. Furthermore, migrasomes 500–3000 nm, have been recently characterized as a type of EV formed from the retraction fibers and released to the periplasmic matrix during cell migration. They carry tetraspanin 4 (TSPAN4) as a marker to their subtype [35]. More newly identified subtypes of EVs have been reported: exophers 1000–10,000 nm, possess markers such as microtubule-associated protein 1A/1B-light chain 3 (LC3), and mitochondrial import receptor subunit 20 (TOM20) [36]. Expophers are yet to be studied in humans. Lastly, exomeres < 50 nm, approximately 35 nm, have been recently described as non-membranous particles [37]. These particles, with the other EVs and particles derived from tumors, have been reported to induce the re-metastatic niches in the formation and pathological effects of organs. Furthermore, they can promote inflammation and downregulation of lipid catabolism in non-tumor livers [38].

## 2. Function of Extracellular Vesicles

### 2.1. Extracellular Vesicles in Bacterial Lung Infections

Host EVs that are released by the host have been thoroughly investigated in various bacterial lung infections. In tuberculosis, the isolated EVs from the cells infected by *Mycobacterium tuberculosis* (Mtb) promote the recruitment and activation of immune cells. Thus, they contribute to the innate immune response. Furthermore, EVs have been suggested to be significant for antigen presentation in Mtb-infected mice, where the infected Siglec-1 knockout mice showed higher apparency in the local spread of bacteria compared to the wild-type mice despite the same bacterial load, suggesting that Siglic-1 is important to inducing antigen presentation, which is mediated by EVs [39]. Immunologically, EVs from Mtb-infected macrophages were also able to activate the endothelial cells, promoting cell migration through the cell monolayer. This also includes the upregulation of genes such as *ccl2*, *vcam 1*, and *cxc1*, cytokine–cytokine receptor interaction, and increased levels of chemokines and adhesion molecules, e.g., chemokine (C-C motif) ligand 2 (CCL2) and VCAM1 [40]. It has been found that exosomes derived from THP-1 cells infected by *Mycobacterium bovis* (BCG) can initiate a proinflammatory response in mice with a robust production of IL-12p40 and tumor necrosis factor alpha (TNF-α) in the lung [41]. On the other hand, evaluating the effects of serum EVs from Mtb-infected patients on the viability of THP-1 monocytes and the human peripheral blood mononuclear cells (PBMCs) showed that infecting both cell types with patient serum EVs, led to an increased death rate [42]. Moreover, a study found that treating Mtb-infected macrophages with EVs produced by Mtb-infected neutrophils increases the production of superoxide anion and autophagy in macrophages leading to higher bacterial killing, as higher levels of proinflammatory cytokines (TNF-α, IL-6, and IL-10) were observed in macrophages stimulated with EVs isolated from Mtb-infected neutrophils [43]. Additionally, it was demonstrated that Mtb RNA is transferred in the macrophage-derived EVs through a SecA2-dependent pathway. These EVs cause the induction of IFN response in macrophages, which is based on the foreign-RNA-sensing pathway retinoic-acid-inducible gene I (RIG-I)/MAVS/TBK1/IRF3 [44]. The activation of this pathway is needed for the EVs produced by Mtb-infected macrophages to generate Mtb replication restriction in the surrounding macrophages [44]. Another study reported that EVs isolated from infected J774A.1 macrophages were larger than spontaneously released EVs (S-EV) from the same uninfected cell line. In addition, Mtb-EV was observed to contain *M. tuberculosis* antigens. The study also showed that S-EV reduced the bacterial load and production level of both MCP-1 and TNF-α in *M. tuberculosis*-infected macrophages. Moreover, treating mice infected with *M. tuberculosis* with both S-EV and Mtb-EV led to a significant reduction of bacterial load in the lungs, but no effect was recorded regarding the survival rate or the lung pneumonic area [45].

On the aspect of adaptive immune response, its activation can also be improved by EVs from Mtb-infected cells. It has been shown that the EVs from macrophages infected by *Mycobacterium tuberculosis* and *Mycobacterium bovis* (BCG) can activate the T cells by presenting peptide-MHC-II complexes [46]. Treating mice with EVs from BCG-infected macrophages leads to antigen-specific activation of CD4^+^ and CD8^+^ T cells in the spleen, lung, and mediastinal nodes [47]. 

On the contrary, immune suppression was observed upon treatment with EVs from mycobacterial-infected cells. The expression of macrophages MHC-II and CD4 in response to IFN-γ was partially inhibited upon treatment with EVs produced by Mtb-infected macrophages, which is dependent on TLR2 [48]. In addition, a similar but stronger inhibition of CD4^+^ T cell activation was observed compared to the inhibition caused by lipoarabinomannan (LAM), a virulence factor associated with Mtb. This led to a reduction of IL-2 production and T-cell proliferation [49].

*Streptococcus pneumoniae* is another species that causes bacterial lung infection called pneumococcal pneumonia. Pneumolysin is a pore-forming toxin produced by *S. pneumoniae* and a main contributor to the inflammatory processes implicated in the aforementioned infection [50]. It has been found that the EVs produced by the neutrophils subjected to pneumolysin can cause direct activation of platelets. This included a significant increase in the percentage of platelets expressing CD62P (a P-selectin present in megakaryocytes) and a higher surface expression of CD62P compared to the platelets incubated with the EVs collected from untreated neutrophils [50]. Another finding reported that naïve platelets incubated with EVs produced by neutrophils treated with pneumolysin showed a higher percentage of cells producing CD62P and increased level of surface expression of the same protein with respect to naïve platelets incubated with EVs isolated from non-pneumolysin-challenged neutrophils [50]. An additional study showed that microvesicles isolated from lung epithelial cells stimulated with pneumolysin were able to impair the ROS production in neutrophils, thus suppressing these neutrophils’ defensive response [51]. 

*Staphylococcus aureus* can cause real life-threatening infections in humans. In some cases, it can lead to sepsis and both pneumonia and bone and joint infections. It has been shown that EVs produced by neutrophils treated with *S. aureus* (termed bEVs) can be associated with the exogenously added bacteria causing aggregation [52]. In accordance with that, another study revealed that bEVs were able to induce the proinflammatory response in macrophages exhibited by producing both IL-6 and IL-1β in a dose-dependent manner [53].

*Pseudomonas aeruginosa* is a pathogen that has been reported to form antibiotic-resistant biofilms. Interestingly, a study demonstrated that miRNA (let-7b-5b) transferred by EVs secreted by the airway epithelial cells can act as an RNA interference to reduce the ability of *P. aeruginosa* to form biofilms and increase the sensitivity of this pathogen to beta-lactam antibiotic aztreonam [54].

*Legionella peneumophila* is a bacterial agent that causes severe pneumonia in humans. EVs collected from supernatants of *L. pneumophila*-infected THP-1 were used to infect both THP-1 and A549 healthy cells. The results demonstrated a higher response in THP-1 cells in CXCL8, TNF-α, IL-1β, and MCP-1 expression compared to A549 cells [55] (Figure 2).

### 2.2. Extracellular Vesicles in Viral Lung Infections

The influenza virus is responsible for one of the major annual seasonal epidemics and pandemics. It infects the nasal and tracheal airways and later spreads among the upper and lower respiratory tract [56]. It has been demonstrated that miR-17-5p and miRNA, which are highly expressed in lung epithelial cells infected by influenza A virus (IAV) and present in bronchoalveolar lavage fluid (BALF), can lead to the expression reduction of the antiviral factor Mxl, which in turn enhances virus replication [57]. On the contrary, EVs released from influenza-virus-infected lung cells can be a player in the antiviral response. The released EVs that include functional viral RNAs (e.g., mRNA and miRNA) can be engulfed by dendritic cells. Then, a large quantity of type I interferon and proinflammatory cytokines (IL-12 p40 and IL-6) is produced upon the internalization of these RNAs and their recognition by pattern recognition receptors (PRRs). For instance, TLR7 can recognize IVA-RNA in plasmacytoid dendritic cells, generating the induction of the antiviral innate immune responses signal and thus type I interferon production [58]. Moreover, EVs produced by infected epithelial cells are shown to be able to control the antiviral responses of the neighboring epithelial cells and immune cells. An activation of hsa-miR-1975, a Y5-RNA-derived small RNA, takes place in the apoptosis process in human lung adenocarcinoma epithelial A549 cells infected with the influenza virus [59]. This hsa-miR-1975 is delivered into the surrounding neighboring cells by exosomes, where it fuses with other antiviral proteins or nucleotides to produce interferon, thereby blocking viral replication when it invades the cells [59,60]. In addition, exosomes isolated from A549-infected influenza showed 73 overexpressed and 24 underexpressed miRNAs compared to uninfected cells, showing a significant upregulation of five miRNAs (hsa-miR-572, hsa-miR-141#, hsa-miR-196b, hsa-miR-24-2#, hsa-miR-483-3p), and significant downregulation of five miRNAs (hsa-miR-194, hsa-let-7d, hsa-miR-361, hsa-miR-223, hsa-miR-671-3p) [61].

Additionally, respiratory cells produce EVs that can induce an antiviral response. For instance, exosome-like vesicles (30–100 nm) carrying multivesicular and late endosomal membrane markers Tsg101 and CD63 were shown to neutralize influenza viruses with the help of α-2,6 linked sialic acid presented on their surface, which binds preferentially to the influenza viruses, allowing for a mucosal innate response regulated by these EVs [62]. Another study showed that the BALF of influenza-virus-infected mice contained an increased quantity of exosomes enriched with miR-483-3p. These exosomes are believed to mainly derive from macrophages but not other lung tissue cells. Using these exosomes to transfect epithelial cells in the lung, leads to the expression of type I interferon and proinflammatory cytokines, such as interleukin 6 (IL-6), (CCL2), TNF-α, and SP110, where miR-483-3p targets negative regulators of the RIG-I signaling pathway [63]. Furthermore, in a consecutive study, exosomes rich in miR-483-3p were also discovered in the serum of mice infected with influenza virus together with high levels of inflammatory cytokines in vascular endothelial cells [64]. In addition, exosome production can be coordinately involved with autophagy to induce M1 macrophage polarization and macrophage recruitment. Moreover, it has been shown that this recruitment and polarization of M1, during H1N1 infection, is activated through an autophagy-exosome-dependent pathway [65].

Additionally, immune cells such as alveolar macrophages (AMs) play a role in host defense against infection through their released EVs. It has been demonstrated that the EVs released from naïve rodent and human macrophages are engulfed by the alveolar epithelial cell’s endosomes, leading to the acceleration of endosomal acidification, inhibiting the fusion of the influenza virus and thus preventing the virus’s nuclear entry and therefore its replication [66]. 

Respiratory syncytial virus (RSV) is one of the most common respiratory viruses causing cold and respiratory symptoms [67]. RSV is harmless in most cases unless it affects early born babies and infants [68]. RSV is the most common cause of bronchiolitis and pneumonia in children younger than 1 year of age [67,68]. Exosome-wise, it was identified that in RSV infection, the pattern of exosomal (secretome) proteins from epithelial cells changes drastically. For instance, CXCL10 and CCL5 are upregulated in the epithelial cell exosomes in RSV infection compared to healthy cells [69]. CXCL10 plays an important role as a modulator of the innate and adaptive immune response and as a cell growth regulator and also possesses antagonistic effects [70], while CCL5 recruits CD4^+^ and CD8^+^ T cells and has an essential role in controlling viral replication [71]. Moreover, exosomes derived from epithelial cells infected with RSV contain different components of the virus [72]. These virus-induced exosomes have the potential to induce proinflammatory signals in both monocytes and epithelial cells. Monocytes transfected with exosomes isolated from virally infected cells were able to cause upregulation in the production of both monocyte chemoattractant protein-1 (MCP-1) and interferon-gamma-induced protein 10 (IP-10) and regulated normal T cell expression secretion upon activation (RANTES). On the other hand, A549 epithelial cells were able to produce tumor necrosis factor alpha (TNF-α, in addition to the up-mentioned cytokines) [72]. In addition, exosomes released from the infected epithelial cells stimulate monocytes to produce chemokines, and these exosomes were not able to transmit the RSV infection to healthy cells. Additionally, the exosomal cargo from RSV-infected cells shows an increased proportion of small regulatory RNAs besides the content of viral proteins and RNA [72]. This shows that RSV-infected cells are able to produce various types of exosomes, leading to differential proinflammatory stimulation of the immune system.

In a study on asthmatic patients, the authors investigated a longitudinal analysis of the exosomal miRNAs (ExoMiRNAs) of the circulating exosomes before and after the infection with rhinovirus. Following the infection with rhinovirus, the authors divided the studied miRNAs into two clusters: an upregulated cluster with its miRNA upregulated in asthmatic subjects compared to healthy subjects after the initial downregulation (e.g., hsa-let-7f-5p, hsa-let-7a-5p, and hsa-miR-30c-5p) and a downregulated cluster with its miRNA downregulated in the asthmatic subjects compared to healthy subjects after the initial upregulation (e.g., hsa-miR-122-5p, hsa-miR-101-3p, hsa-miR-423-5p) [69,73]. Meanwhile, the upregulated cluster miRNAs were shown to strongly correlate with Th1- and interferon-induced cytokines/chemokines (IFN-γ and IP-10), and the downregulated cluster miRNAs were significantly correlated with Th2 (IL-13), expressing their probable roles in Th2 immunity [73]. Another study on asthmatic patients investigated the relationship between rhinovirus infection of the bronchial epithelial cells and the release of tenascin-C (TN-C) and small EVs, such as exosomes and small microvesicles, after stimulation with TLR3 agonist and poly(I:C) as a synthetic viral mimic. This study also investigated the function of the released proteins/vesicles. This study showed that poly(I:C) stimulation and rhinovirus infection of asthmatic PBECs (primary human bronchial epithelial cells) increased the release of T-NC in vitro [74]. Moreover, TN-C and small EVs secreted from BEAS-2B cells simulated with poly(I:C), were able to promote cytokine synthesis in macrophages and BEAS-2B cells, but small EVs secreted from control cells did not [74].

Adenoviruses are very well described in the literature for their role as respiratory pathogens and as tools for gene delivery for genetic studies and therapy. However, a study investigated the ability of exosomes to mediate the viral cell entry, discovered that exosomes released from neural stem cells were able to mediate the cellular entry for adenovirus type 5 in a receptor-independent manner, leading to the infection of new healthy cells. On the other hand, this activity was depleted by treating the exosomes with specific antibodies against T-cell immunoglobulin mucin protein 4 (TIM-4) [75]. Studying the role of adenovirus type 3 in regulating the biogenesis and composition of EVs released by A549 infected cells, a study revealed a significant increase in mean particle size, concentrations, and total EV protein content at a higher increase in multiplicity of infections (MOIs). Moreover, significantly elevated levels of Rab5, 7, and 35, which are considered Rab GTPases, correspond to the increase in MOIs compared to controls, suggesting that these modulations could impact the infection pathogenesis and disease progression [76] (Figure 2).

In 2020, SARS-CoV-2 virus caused the COVID-19 global pandemic. A major part of the research in the past three years has been focused on the pathogenicity, immunity, prevention, and treatment of COVID-19 infection. Many studies have investigated the involvement of EVs, mainly exosomes, in COVID-19 infection. Exosomes are upregulated in PBMCs in severe SARS-CoV-2 infection patients [77]. They lead to the upregulation of different proinflammatory cytokines, including IL-6, IL-8, and TNF-α [78]. Furthermore, SARS-CoV-2 spike protein was able to induce immunologically active EVs upon treatment of dendritic cells which led to changes in both CD4^+^ and CD8^+^ cells [79]. These EVs also had high concentrations of neutrophil elastase (NE), which was correlated with endothelial damage in COVID-19 patients [80] and further triggered NLRP3 inflammasome in endothelial cells [81]. In addition to that, the aldo-keto reductase 1B10 (AKR1B10) enzyme transferred via EVs to different immune cells, triggering a cytokine storm in COVID-19 infection [82]. Another content of EVs isolated from COVID-19 patients’ serum is miRNAs. miR-3168 was upregulated in COVID-19 pneumonia patients but significantly downregulated in COVID-19 acute respiratory distress syndrome (ARDS) compared to healthy control. These were correlated to lower interleukin-8 [83]. Angiotensin-converting enzyme 2 (ACE2)^+^ exosomes induced by COVID-19 infection could bind to viral particles and block viral entry, suggesting an active defensive role for those EVs [84,85,86]. Furthermore, circulating CD62E^+^ endothelial EVs were positively correlated with hospital mortality [87]. Moreover, platelet-derived EVs in hospitalized COVID-19 infection patients were associated with worse prognosis and mortality [88]. On another note, these EVs have the ability to induce lipid raft formation in epithelial cells in human small airways [89]. Furthermore, these EVs could contain a large number of active virus particles, which plays an important role in disease transmission [90,91]. These epithelial-derived EVs also mediate cardiac inflammation through encapsulated surfactant protein C [92]. Interestingly, EVs shed from virus-infected cells had the potential to activate thrombin generation and thus cause coagulation [93,94,95,96,97]. This is regulated by exosomal miR-145 and miR-885 [98]. Moreover, it was revealed by Kaur et al. that the upregulation of cytokine signaling in platelet-derived EVs is associated with thrombophilia [99]. On the other side, EVs released by SARS-CoV-2-infected cells can transfer viral particles and other proteins to neurons, astrocytes, and microglia, leading to a fastened fibril formation and progression of neurological symptoms related to COVID-19 infection [100,101]. In addition, it has been found that exosomes isolated from lungs infected with SARS-Cov-2 can transport mRNAs encoding for several transcription factors (LITAF, IRF2, IRF9, PHF11, ZNF385A, MIER1, SP140L, BCL3, STAT4, NFKBID, TRIM22, JUND, STAT1, BLOC1S1, SP110, TRIM38, MXD1, SP140, and HESX1), crossing the blood–brain barrier, suggesting contributions of these transcription factors in neurodegenerative processes upon their translation. These contributions are inflammation, apoptosis, and other signal transduction processes [102] (Figure 3).

Lastly, exosomes isolated from lung transplant recipients showing symptoms of lower- and upper-tract respiratory viral infections such as rhinovirus, and coronavirus carry lung-self antigens, 20S proteasome, and viral antigens. These exosomes induced immune responses against lung self-antigens, which led to chronic lung allograft dysfunction in mice immunized with the aforementioned exosomes [103].

### 2.3. Extracellular Vesicles as a Biomarker in Lung infections

Besides the role of extracellular vesicles in the modulation of immune responses, they can be used as a biomarker for various lung infections. In tuberculosis, exosomal miRNA-185-5p, ASdes, and MTB-miR5 can be used as novel biomarkers [104,105]. Heat shock protein (Hsp16.3) encapsulated in exosomes can be used as a biomarker for *Mycobacterium tuberculosis* infection [106]. In pneumococcal infections, EVs produced bear a high potential to identify multiple biomarkers. Phosphatidylserine and microRNAs in EVs of infected patients were used to discriminate between survivors and non-survivors in pneumonia [107,108]. Virus-wise, exosomal microRNAs from adenovirus pneumonia infection can be used as a diagnostic biomarker in the serum of infected children [109]. Moreover CD13^+^ and CD82^+^ EVs from COVID-19 patients can predict the severity of the disease, while CD24^+^ EVs are more abundant in mild COVID-19 and healthy donors [110]. In addition to that, multiple EV proteins have been identified to act as early predictors of COVID-19 severity, including coat complex subunit beta 2 (COPB2), KRAS proto-oncogene (KRAS), protein kinase C beta (PRKCB), and ras homolog family member C (RHOC) [111].

## 3. Conclusions

In conclusion, in this review article, we have gathered all the known knowledge about the role of EVs in the pathogenesis of various bacterial and viral lung infections. EVs can play a pivotal role in the pathogenesis of various lung infections. They regulate the immune system on one side and lead to changes in structural cells in the lung on the other side. Furthermore, EVs and their cargo can be used as biomarkers for disease severity and progression. In bacterial infections, EVs have been shown to lead to multiple immunological processes, and bacteria use them to transfer material and information to other cells. Moreover, viruses, including the SARS-CoV-2 virus, hijack EVs to transfer their material into new cells to exert an undesired immune response. In addition to that, EVs hold a very high potential in their immune-regulatory effects, and they can be one of the main keys to understanding various pathomechanisms in both bacterial and lung infections (Table 1). Much more work on EVs in lung infections is needed to understand and distinguish between various types of EVs, their role, and their origin.

## Figures and Tables

**Figure 1 ijms-24-16139-f001:**
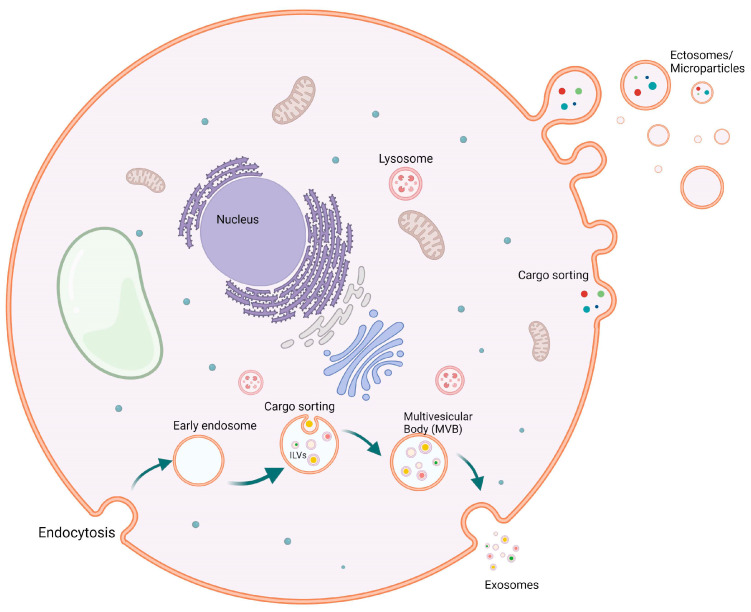
Biogenesis of extracellular vesicles (EVs). Biogenesis of both exosomes (50–150 mm) and ectosomes (50–10,000 mm) from healthy cells. Exosomes are budding from early endosomes after cargo sorting, where multivesicular bodies are formed, which then give rise to exosomes. On the other hand, ectosomes or microparticles are bigger particles that are budded through the cell membrane carrying various cargos.

**Figure 2 ijms-24-16139-f002:**
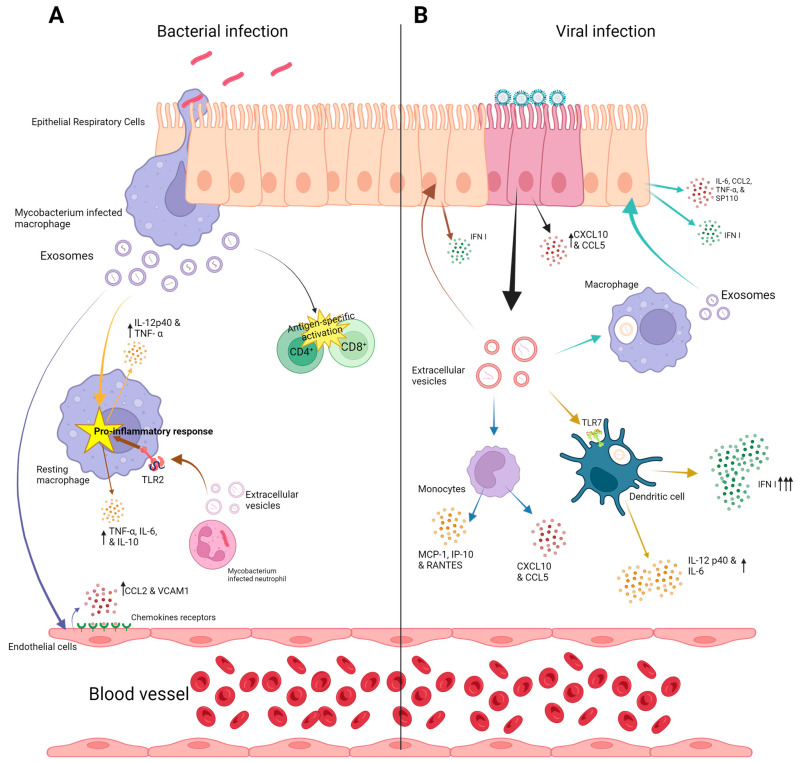
Exosome function in lung infections (**A**); bacterial lung infection: In bacterial lung infections, epithelial cells, macrophages, and endothelial cells play an important role in pathogenesis through their exosomal cargo. Various cytokines and chemokine production due to exosome effects are shown. (**B**) Viral lung infection: The contribution of exosomal cargo from macrophages, epithelial cells, monocytes, and dendritic cells is depicted. Various cytokines and chemokine production due to exosome effects are shown. Upward arrows indicate increased production of various chemokines and cytokines.

**Figure 3 ijms-24-16139-f003:**
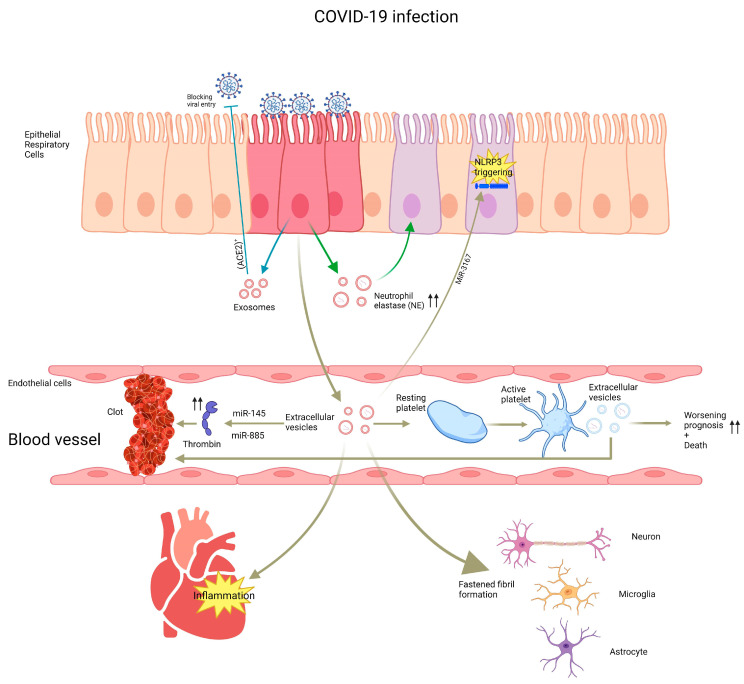
Contribution of exosomes in COVID-19 infection. Exosomes produced due to COVID-19 infection and their role in the pathogenesis of the infection. Epithelial exosomes can cause blockade of entry and triggering of inflammasome. Platelet- and serum-derived exosomes can lead to the activation of thrombin, activation of the fibril in the nervous system, and inflammation of the heart. Upward arrows indicate increased production of various chemokines and cytokines.

**Table 1 ijms-24-16139-t001:** Summary of Bacterial and Viral Lung infection related EV effects.

Pathogens	Effect	References
*Mycobacterium tuberculosis*	Infected Mφ EVs: + (CCL2, VCAM1) in endothelial cells ++ Peptide-MHC-II complexes in T-cells Serum EVs: ++ Death rate in THP-1 & PBMCs Infected neutrophils EVs: ++ Mφ autophagy ++ Mφ superoxide anion ++ Mφ bacterial killing ++ (TNF-α, IL-6, IL-10) in Mφ Mtb RNA in EVs: ++ IFN in Mφ Spontaneously released EVs: -- MCP-1, -- TNF-α, -- bacterial load in mtb-infected Mφ Mtb-EVs & S-EVs: --- Bacterial load in Mtb-infected mice.	[39,41,42,43,45]
*Mycobacterium bovis*	Infected THP-1 EVs: +++ (IL-12p40, TNF) Infected Mφ EVs ++ Peptide-MHC-II complexes in T-cells + Antigen-specific activation of CD4^+^ & CD8^+^ T cells in mice	[40,45,46]
*Streptococcus pneumoniae*	Pneumolysin+ Neutrophils EVs: ++ CD62P in platelets Pneumolysin+ Lung epithelial cells microvesicles: -- ROS in neutrophils	[49,50]
*Staphylococcus aureus*	Infected neutrophils EVs: ++ (IL-6, IL-1β) in Mφ	[52]
*Pseudomonas aeruginosa*	Let-7b-gb miRNA in EVs: - Biofilm formation, + sensitivity to beta-lactam antibiotic (Aztreonam)	[53]
*Legionella penuemophila*	Supernatant EVs: ++ (CXCL8, TNF-α, IL-1β, MCP-1) in THP-1	[54]
Influenza A	Infected lung cells EVs: +++ (Type I IFN, IL-12p40, IL-6) in dendritic cells Blocking viral replication in A549 infected cells Infected A549 cells EVs: Neutralizing virus particles Infected lung Mφ EVs: ++ (Type I IFN, IL-6, CCL2, TNF-α) in epithelial cells Alveolar macrophages EVs: Acceleration of endosomal acidification in epithelial cells	[57,58,59,61,62,65]
Respiratory syncytial virus	Infected Epithelial cells exosomes showed +(CXCL10 and CCL5). ++ (MCP-1, IP-10 & RANTES) in monocytes & A549 cells ++ TNF-α in A549 cells	[68,71]
Rhinovirus	+ (IFN-γ & IP-10) in Th1 + IL-13 in Th2	[72]
Adenovirus	Adenovirus 3 EVs: +++ Rab5, 7 & 35	[75]
SARS-CoV-2	+ (IL-6, IL-8 and TNF-α) ++ neurtrophil elastase-related endothelial damage Aldo-Keto Reductase 1B10-related cytokine storm - IL-8 (ACE2)^+^-related viral entry block CD62E^+^-related hospital mortality Platelets EVs-related worsened prognosis and mortality Lipid raft formation in human small airways Active viral particles-related disease transmission Epithelial EVS-related cardiac inflammation Coagulation Platelet-derived EVs cytokine signaling upregulation-related thrombophilia Fastened fibril formation and progression in neurons, astrocytes and microglia Transcription factors transmission-related neurodegenerative processes	[77,79,81,83,84,85,86,87,88,89,90,91,92,93,94,95,96,98,99,100,101]

+ upregulation, ++ high upregulation, +++ Robust upregulation, - downregulation, -- downregulation/inhibition, --- total inhibition.

## Data Availability

There is no data contained in this manuscript.

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
