# Peer review of "Extracellular Vesicles and Their Role in Lung Infections"

_ijms, 2023, doi:10.3390/ijms242216139_

Round 1

Reviewer 1 Report (Previous Reviewer 1)

Comments and Suggestions for Authors

The present review manuscript authored by Hambo et al. is a de novo submission of a previously submitted manuscript. Despite efforts of the authors, the following previous raised comments have not been satisfactorily covered:

-          In the abstract authors should highlight the novelty of the manuscript over published literature.

-          Statement of line 124 “Host EVs that are released by the host have been thoroughly investigated in various bacterial lung infections.” does not tackle the issue regarding phosphatidylserine as stated in the author’s reply.

-          Authors claimed that arguing about EVs derived from mesenchymal stem cells as therapeutic agents is beyond the scope of the review. Therefore, the section entitled “Extracellular Vesicles as a Biomarker and Therapeutic Target” has no sense. Of note, only lines 346-350 addressed the therapeutic potential of EVs showing only two studies. The section title should be changed or the section expanded.

-          A table summarising all the studies on bacterial and virus lung infections is still not present. Only a citing text as (table 1) in the conclusion section (line 361).

-          Although some parts of the text have been rewritten, the manuscript still presents a high number of spelling and grammatical mistakes that require due revision.

-          Other minor comments such as distinct font sizes have been used throughout the text such as in page 5. The amendment of has-miRNA-1975 has been adequately carried out but not in line 186 as stated by the authors.

Comments on the Quality of English Language

Although some parts of the text have been rewritten, the manuscript still presents a high number of spelling and grammatical mistakes that require due revision.

Author Response

We thank the reviewer for his comments, we have addressed all the points one by one as follows

  • In the abstract authors should highlight the novelty of the manuscript over published literature.

We have added couple of sentences to highlight the novelty of the manuscript-

  • Statement of line 124 “Host EVs that are released by the host have been thoroughly investigated in various bacterial lung infections.” does not tackle the issue regarding phosphatidylserine as stated in the author’s reply.

We apologize, due to changes in the manuscript, the lines shifted, the statement of phosphatidylserine can be found in lines 119-121 and 383-385

  • Authors claimed that arguing about EVs derived from mesenchymal stem cells as therapeutic agents is beyond the scope of the review. Therefore, the section entitled “Extracellular Vesicles as a Biomarker and Therapeutic Target” has no sense. Of note, only lines 346-350 addressed the therapeutic potential of EVs showing only two studies. The section title should be changed or the section expanded.

The reviewer is correct, we have changed this section entirely

  • A table summarising all the studies on bacterial and virus lung infections is still not present. Only a citing text as (table 1) in the conclusion section (line 361).

The table is now attached to the main document and uploaded as Table1 in the submission system

  • Although some parts of the text have been rewritten, the manuscript still presents a high number of spelling and grammatical mistakes that require due revision.

We have revised the manuscript entirely for spelling and grammatical mistakes

  • Other minor comments such as distinct font sizes have been used throughout the text such as in page 5. The amendment of has-miRNA-1975 has been adequately carried out but not in line 186 as stated by the authors.

We have revised the manuscript, the references attached and made sure that all hsa-miR-1975 is written correctly.

Reviewer 2 Report (New Reviewer)

Comments and Suggestions for Authors

In the manuscript “Extracellular vesicles and their role in lung infections” the authors collected the knowledge about the role of EV in bacterial as well as viral lung infections. This review appears to be a valuable study on the subject, collects a lot of literature data in one place, and will be helpful to researchers and medical doctors in understanding the role of EVs in lung infections. The literature review includes current writing including eight papers from 2023.

The review is oriented on host-derived EV after pathogen infections. From another point of view i.e. from the pathogen perspective this subject will be quite different but also interesting i.e. how bacterial/virus-derived EV influences pathogenesis.

In EVs classification should be mentioned that bacterial EVs also have different names at least mention OMV. Also, according to my knowledge, the surface markers used to differentiate ectosomes are not found in bacterial EVs. So the author should clearly state their focus on eucaryotic EVs (host-derived EVs), especially since the first statement in the Introduction suggests that it concerns also procaryotic EVs.

Minor remarks

In general manuscript is well prepared and I have only some minor comments to improve the quality of the manuscript.

Abbreviations

The abbreviations should be placed in the alphabetic order.

p.2. l. 32 MAP Mycobacterium avium subsp. paratuberculosis

p.2 l. 36 TN-C: Tenascin-C

p.2, l. 52           Hsp: Heat shock protein

Please add SP110, RIG-I, RSV, and RANTES to the abbreviations list.

p.5 l. 113, please remove “has”

p.5. l. 115, please correct “Exophers”

p.5 l.117-119 please change the font

p.6. l.139 please correct “Siglec-1”

p.8 l.193 please correct “Legionella pneumophila

p.8 l. 199 please correct “The influenza virus is responsible for one of the major annual seasonal epidemics and pandemics”

p.10 l. 235 please change from “in H1N1 infection” to “during H1N1 infection”

p. 10 l. 250 please change to the small letter “exosomes”

p.11. l. 284 please replace from “adenovirus 5” to “adenovirus type 5” for clarity

p.11. l. 287 please replace from “adenovirus 3” to “adenovirus type 3” for clarity

p.12. l. 290 please explain Rab 5,7 and 35 as “ Rab GTPases”

p. 12 from 293 to 327 please use the uniform nomenclature for COVID-19

p.12 l. 310 “Platelet-derived EVs” from small letter

p.13. l.319 “ability transfer” change to “ability to transfer”

p.13 l. 334 “Extracellular” from small letter

p.14. l. 351 “conclusion” please remove the dot

p.14. l. 356-358 please change to “In bacterial infections, EVs have been shown to lead to multiple immunological processes and bacteria use them to transfer material and information to other cells”.

p. 14. l. 363 please change “various types of EVs their role and their origin” or “various types of EVs and their role as well their origin”

Author Response

We thank the reviewer for his comments. We have addressed all the comments point by point and worked on the English grammer of the manuscript

  • The review is oriented on host-derived EV after pathogen infections. From another point of view i.e. from the pathogen perspective this subject will be quite different but also interesting i.e. how bacterial/virus-derived EV influences pathogenesis.

We thank the reviewer for his comment. We agree entirely regarding this point. We decided to focus on cell-derived EVs in the context of lung infections. Bacterial and Virus derived EVs or loaded EVs is a much-complicated subject that even microbiota can contribute to it and we decided to leave it for a later timepoint to be tackled.

  • In EVs classification should be mentioned that bacterial EVs also have different names at least mention OMV. Also, according to my knowledge, the surface markers used to differentiate ectosomes are not found in bacterial EVs. So the author should clearly state their focus on eucaryotic EVs (host-derived EVs), especially since the first statement in the Introduction suggests that it concerns also procaryotic EVs.

We have mentioned that we are focusing on eukaryotic EVs in both the abstract and introduction. Furthermore, we add 2 sentences about the naming of bacterial EVs at the end of introduction.

Minor remarks

In general manuscript is well prepared and I have only some minor comments to improve the quality of the manuscript.

Abbreviations

  • The abbreviations should be placed in the alphabetic order.

We have arranged all abbreviations in alphabetical order

  • 2. l. 32 MAP Mycobacterium avium subsp. Paratuberculosis

Corrected

  • 2 l. 36 TN-C: Tenascin-C

Corrected

  • 2, l. 52 Hsp: Heat shock protein

Corrected

  • Please add SP110, RIG-I, RSV, and RANTES to the abbreviations list.

The new abbreviations have been added

  • 5 l. 113, please remove “has”

Corrected

  • 5. l. 115, please correct “Exophers”

Corrected

  • 5 l.117-119 please change the font

Corrected

  • 6. l.139 please correct “Siglec-1”

Corrected

  • 8 l.193 please correct “Legionella pneumophila”

Corrected

  • 8 l. 199 please correct “The influenza virus is responsible for one of the major annual seasonal epidemics and pandemics”

Corrected

  • 10 l. 235 please change from “in H1N1 infection” to “during H1N1 infection”

Corrected

  • 10 l. 250 please change to the small letter “exosomes”

Corrected

  • 11. l. 284 please replace from “adenovirus 5” to “adenovirus type 5” for clarity

Corrected

  • 11. l. 287 please replace from “adenovirus 3” to “adenovirus type 3” for clarity

Corrected

  • 12. l. 290 please explain Rab 5,7 and 35 as “ Rab GTPases”

Corrected

  • 12 from 293 to 327 please use the uniform nomenclature for COVID-19

Corrected and unified

  • 12 l. 310 “Platelet-derived EVs” from small letter

Corrected

  • 13. l.319 “ability transfer” change to “ability to transfer”

Corrected

  • 13 l. 334 “Extracellular” from small letter

Corrected

  • 14. l. 351 “conclusion” please remove the dot

Corrected

  • 14. l. 356-358 please change to “In bacterial infections, EVs have been shown to lead to multiple immunological processes and bacteria use them to transfer material and information to other cells”.

Corrected

  • 14. l. 363 please change “various types of EVs their role and their origin” or “various types of EVs and their role as well their origin”

Corrected

Round 2

Reviewer 1 Report (Previous Reviewer 1)

Comments and Suggestions for Authors

No further comments

Comments on the Quality of English Language

No further comments

This manuscript is a resubmission of an earlier submission. The following is a list of the peer review reports and author responses from that submission.

Round 1

Reviewer 1 Report

Comments and Suggestions for Authors

The present review manuscript authored by Hambo et al. focuses on the role of extracellular vesicles (EVs) in bacterial and viral lung infections, specifically to their therapeutic and diagnostic functions. While figures are clear and comprehensive, the manuscript is incomplete and does not cover all relevant literature in the field. I have the following comments:

-          Abstract provides insufficient information.

-          Background section regarding EVs is outdated and state-of-the-art references in the field should be added.

-          Currently, the classification of EVs is under debate since exosomes, microvesicles and apoptotic bodies overlap in some physical and chemical properties.

-          Microvesicles do contain mitochondrial proteins, please see PMID: 33785738 (Line 115).

-          Phosphatidylserine can also be found and is a marker of microvesicles not only apoptotic bodies (Line 124).

-          COVID-19 infection should be a subsection within the above section about Viral Lung Infections.

-          Important studies regarding EVs and lung infections are overlooked.

-          Tackling EVs derived from mesenchymal stem cells as therapeutic agents is missing.

-          Figure 1 is misleading: exosomes, microvesicles and apoptotic bodies can indistinctively be released from either healthy or diseased cells; activated, non-activated and apoptotic cells, under distinct types of stimuli.

Minor comments

-          Avoid using the same wording. E.g.: ‘attributing’ and ‘attributed’ (Lines 58-59).

-          It would be desirable to add a table summarising studies on bacterial and virus lung infections.

-          ‘hsa-miRNA-1975’ instead of ‘has-miRNA-1975’ (Line 186).

Comments on the Quality of English Language

Multiple English grammatical errors. For instance: ‘revisit’ instead of ‘visit’ (Line 69), ‘are’ should be added between ‘which’ and ‘released’ (Line 69), among others.

Reviewer 2 Report

Comments and Suggestions for Authors

The review article written by Hambo et al., has collectively summarized the recent findings on the role of exosomes in bacterial and lung infections. However, it requires a substantial amount of spelling and language correction throughout the paper. I have listed some of them below but could not cover it all.

1.      Line 35; Needs to be corrected for Tumor Necrosis Factor Alpha and TNFa.

2.      Line 49; Abstract needs to be re-written. Authors use infections and disease condition terminologies interchangeably. It’s not clearly mentioned. Also, the word count in the bracket needs to be removed from the heading. Also, the abstract needs to be a single paragraph without subheadings like Introduction and Conclusion as per the author instructions of the journal. For example, In line 256 its mentioned that “COVID19 Infection Patients” instead of “SARS-CoV-2 infected patients”.  And in line 278, “COVID19 infected cells” instead of “SARS-CoV-2 infected cells”, and so on.

3.      The names of bacteria were not mentioned uniformly throughout the article. It needs to be italicized throughout.

4.      microRNA names need to be correctly used. miR prefix denotes the matured short form of the microRNA while “mir” denotes the precursor of the miRNA.

5.      Line 217; Do the authors mention “Poetical” instead of Potential?

6.      Throughout the paper authors need to follow the expansion of a term at the first use and abbreviation thereafter in a uniform way. For example, Line 220; mentions tumor necrosis factor in full while TNF has already been mentioned earlier.   

7.      The last statement in the conclusion is not clear and needs to be re-written by including “both bacterial and viral lung infections”.

8.      Figure 2; labelling for A and B is missing on the figure.

9.      Figure 3: Heading needs to be changed to “SARS-CoV-2 Infection” instead of “COVID Infection”.

Comments on the Quality of English Language

Needs to be re-written to correct several misleading spelling and language mistakes.